# Redox-Responsive Lipidic Prodrug Nano-Delivery System Improves Antitumor Effect of Curcumin Derivative C210

**DOI:** 10.3390/pharmaceutics15051546

**Published:** 2023-05-19

**Authors:** Xin Guo, Min Wu, Yanping Deng, Yan Liu, Yanpeng Liu, Jianhua Xu

**Affiliations:** 1The School of Pharmacy, Fujian Medical University, Fuzhou 350122, China; 17705005241@163.com (X.G.); 18450056606@163.com (M.W.); 13959147061@163.com (Y.L.); 15617186956@163.com (Y.L.); 2Fujian Provincial Key Laboratory of Natural Medicine Pharmacology, Fujian Medical University, Fuzhou 350122, China

**Keywords:** curcumin derivatives, redox-responsive, self-assemble, prodrug, bioavailability, antitumor

## Abstract

The poor bioavailability of curcumin and its derivatives limits their antitumor efficacy and clinical translation. Although curcumin derivative C210 has more potent antitumor activity than curcumin, it has a similar deficiency to curcumin. In order to improve its bioavailability and accordingly enhance its antitumor activity in vivo, we developed a redox-responsive lipidic prodrug nano-delivery system of C210. Briefly, we synthesized three conjugates of C210 and oleyl alcohol (OA) via different linkages containing single sulfur/disulfide/carbon bonds and prepared their nanoparticles using a nanoprecipitation method. The prodrugs required only a very small amount of DSPE-PEG2000 as a stabilizer to self-assemble in aqueous solution to form nanoparticles (NPs) with a high drug loading capacity (~50%). Among them, the prodrug (single sulfur bond) nanoparticles (C210-S-OA NPs) were the most sensitive to the intracellular redox level of cancer cells; therefore, they could rapidly release C210 in cancer cells and thus had the strongest cytotoxicity to cancer cells. Furthermore, C210-S-OA NPs exerted a dramatic improvement in its pharmacokinetic behavior; that is, the area under the curve (AUC), mean retention time and accumulation in tumor tissue were 10, 7 and 3 folds that of free C210, respectively. Thus, C210-S-OA NPs exhibited the strongest antitumor activity in vivo than C210 or other prodrug NPs in mouse models of breast cancer and liver cancer. The results demonstrated that the novel prodrug self-assembled redox-responsive nano-delivery platform was able to improve the bioavailability and antitumor activity of curcumin derivative C210, which provides a basis for further clinical applications of curcumin and its derivatives.

## 1. Introduction

Cancers are serious diseases in humans [1,2] and chemotherapy is one of the important strategies for cancer treatment [3]. Chemotherapeutic drugs mainly face several challenges due to high toxicity and drug resistance [4,5]. Therefore, it is essential to find low-toxicity and multi-targeted compounds against cancer.

Curcumin, a polyphenolic compound of natural origin, has been extensively studied and is considered to be a promising lead compound due to a broad antitumor spectrum via multi-targets [6] and its low toxicity [7]. However, its poor bioavailability and unsatisfactory antitumor efficacy in vivo limits its clinical transformation. To overcome the disadvantage of curcumin, our team synthesized a series of curcumin derivatives [8,9,10,11]. Of which, C210 ((1E,6E)-4-(4-hydroxy-3-methoxybenzyl)-1,7-bis(3,4,5-trimethoxy phenyl)hepta-1,6-diene-3,5-dione) has exhibited more potent antitumor activities than curcumin [12]. However, on account of its poor water solubility and quick metabolism, its pharmacokinetic property remains to be improved. Although the application of the formulation technique in the most recent ten years has made up for the deficiency in curcumin derivatives to a certain extent [13], it is not enough to meet the need of effectively improving their antitumor effect in vivo [14,15]. Therefore, it is urgent to find a drug delivery system suitable for C210 with high drug load and high bioavailability.

The prodrug self-assembly nano-delivery system has been widely studied for its ability to improve bioavailability and targeting [16,17,18]. There are many advantages to the nano-delivery system, such as a simple process, high drug loading capacity, less excipients and prolonged systemic circulation time [16,17,18]. Some prodrug formulations based on fatty acids and fatty alcohols could prolong systemic circulation retention time so far as several months or a year [19,20]. In order to enhance tumor selectivity, we need to empower prodrugs to intelligently be released in tumor tissues. Studies have revealed that the cytoplasmic content of glutathione (GSH) and reactive oxygen species (ROS) in tumor cells is about 10 and 100 times higher, respectively, compared to that of in normal cells [21]. Therefore, tumor-environment-responsive (ROS or GSH) prodrug nano-delivery systems have been widely developed [22,23]. Such a drug delivery system was achieved via the redox-responsive prodrug using linkage (e.g., sulfur, selenium bonds and condensed thione) [24,25,26].

In order to deal with the unsatisfactory antitumor effect in vivo due to the low bioavailability of C210, we developed the redox-responsive self-assembled nano-delivery C210 prodrug system. Firstly, we designed and synthesized three conjugates of C210 and oleyl alcohol (OA) via different linkages containing single sulfur/disulfide/carbon bonds (i.e., C210-S-OA, C210-SS-OA and C210-C-OA). The prodrug nanoparticles were prepared using the nano-precipitation method. In this study, the influences of sulfur/carbon bonds on the self-assembly, colloidal stability, redox-responsive drug release, cytotoxicity, pharmacokinetics, biodistribution and antitumor effect of C210 prodrug nanoparticles were systemically investigated.

## 2. Materials and Methods

### 2.1. Materials

2,2’-thiodiacetic acid (98%), 2,2’-dithiodiglycolic acid (96%), glutaric acid (99%), oleyl alcohol (85%), 1-ethyl-3-(3-dimethylaminopropyl) carbodiimide (EDCI, 98%), 4-dimethylaminopyridine (DMAP, 99%), anhydrous sodium sulfate (99%), sodium chloride (99%) hydrogen peroxide (H2O2, 30%) and DL-dithiothreitol (DTT, 99%) were purchased from Aladdin (Shanghai, China). DSPE-PEG2000 (99%) was purchased from Aiweitui Pharmaceutical Technology Co., Ltd. (Shanghai, China). Cyclophosphamide (CTX, 97%) was purchased from Shanghai Yuanye Bio-Technology Co., Ltd. (Shanghai, China). Cell culture media DMEM and RPMI 1640 were purchased from GIBCO (New York, NY, USA). MTT (99%) was purchased from Sigma-Aldrich (Saint Louis, MO, USA). Hoechst 33342 was purchased from Beyotime (Shanghai, China). The TUNEL apoptosis detection kit was purchased from Vazyme (Nanjing, China). Other chemicals and solvents used in this article were of analytical or HPLC grade.

### 2.2. Synthesis and Characterization of C210 Prodrugs

C210-OA prodrugs (C210-S-OA, C210-SS-OA and C210-C-OA) were synthesized via a two-step reaction. Oleyl alcohol (0.268 g, 1 mmol), 2,2’-thiodiacetic acid/2,2’-dithiodiacetic acid/glutaric acid (0.150 g/0.182 g/0.132 g, 1 mmol), EDCI (0.191 g, 1 mmol) and DMAP (0.024 g, 0.2 mmol) were added to 10 mL of dichloromethane solution. The reaction mixture was stirred at room temperature under nitrogen protection. The formation of the target product was detected via TLC. After the reaction, dichloromethane was evaporated, and NaCl saturated brine was added. The aqueous layer product was extracted using dichloromethane, washed with NaCl saturated solution and dried over anhydrous sodium sulfate. The concentrate was separated via silica gel column chromatography to obtain the target product. (E)-2-((2-(octadec-9-en-1-yloxy)-2-oxoethyl)thio) acetic acid (S-OA), (E)-2-((2-(octadec-9-en-1-yloxy)-2-oxoethyl)disulfaneyl) acetic acid (SS-OA) and (E)-5-(octadec-9-en-1-yloxy)-5-oxopentanoic acid (C-OA) were all white oily substances. C210 (0.592 g, 1 mmol), S-OA/SS-OA/C-OA (0.400 g/0.432 g/0.382 g, 1 mmol), EDCI (0.191 g, 1 mmol) and DMAP (0.024 g, 0.2 mmol) were dissolved in 10 mL of water in dichloromethane. The reaction mixture was then stirred at room temperature overnight under nitrogen protection. The formation of the target product was monitored via TLC. After the reaction, the reaction solution was extracted using dichloromethane, washed with NaCl saturated solution and dried over anhydrous sodium sulfate. The concentrate was separated via silica gel column chromatography to obtain the target product. C210-S-OA, C210-SS-OA and C210-C-OA were all yellow oily substances, respectively. Nuclear magnetic resonance (NMR) spectral analyses, infrared spectroscopy (IR) and mass spectrometry (MS) were used to certify the structure of the prodrugs. The purity of the prodrugs was defined via high-performance liquid chromatography (HPLC).

### 2.3. Preparation and Characterization of C210 Prodrug Nanoparticles

Prodrug nanoparticles were prepared using a nanoprecipitation method [25]. A total of 4 mg of prodrugs and 0.8 mg DSPE-PEG2000 were completely dissolved in 1 mL acetone. Then, the mixed solution was added dropwise into 4 mL saline under magnetic stirring, and acetone was removed using a rotary evaporator at 30 °C. Nanoparticles were filtered with a 0.8 μm filter membrane to remove the free prodrugs. To prepare coumarin-6-labeled prodrug nanoparticles, coumarin-6 was co-assembled with prodrugs by injecting the mixture of prodrugs, coumarin-6 and DSPE-PEG2K in acetone into water. Free coumarin-6 was removed using a 0.22 μm filter membrane [26]. The particle size, zeta potential and morphology were measured using a laser particle size analyzer (Anton-Paar, Austria) and transmission electron microscope (FEI, USA). The encapsulation efficiency and drug loading capacity of the prodrug nanoparticles were calculated according to the below Equation (1) and Equation (2), respectively.
Entrapment efficiency (%) = the weight of drug in nanomedicine/the weight of feed drug × 100%(1)
Drug Loading (%) = the weight of drug in nanomedicine/the weight of nanomedicine × 100%(2)

### 2.4. Colloidal Stability

The colloidal stability of prodrug nanoparticles was measured in PBS medium (pH 7.4) containing 10% (*w*/*v*) FBS at 37 °C within 48 h and at 4 °C within 3 months.

### 2.5. Drug release of C210 Prodrug Nanoparticles In Vitro

In vitro release profiles of C210 from the prodrug nanoparticles were measured in PBS (pH 7.4) containing 5% (*w*/*w*) SDS. A total of 1 mL of nanoparticles was incubated in 30 mL of different release medium containing 0, 1 and 10 mM H_2_O_2_ or DL-Dithiothreitol (DTT) at 37 °C [27]. At predetermined time points, 100 μL solution was taken out and replenished with equal release medium. The concentration of the released C210 was determined via HPLC, and the cumulative release rate was calculated using Equation (3). Additionally, after incubating the cells with the prodrug nanoparticles for 24 h, the cells were collected and methanol was added to extract the drug. Additionally, the intracellular concentration of released C210 was determined via LC-MS. The kinetic behavior of the drug release was measured using the zero-order model, first-order model and Higuchi model via Origin 8.0 software.
C210 cumulative released (%) = the released weight of C210/the initial weight of prodrug × 100%(3)

### 2.6. Cell Culture

Human breast cancer MCF-7 cells, mouse breast cancer 4T1 cells, mouse hepatoma H22 cells and human normal breast epithelial cells MCF-10A were bought from the Cell Resource Center, Peking Union Medical College (Beijing, China). Human breast cancer MCF-7 cells and mouse hepatoma H22 cells were cultured in Dulbecco’s Modified Eagle’s Medium (DMEM) with 10% fetal bovine serum (FBS) at 37 °C in 5% CO_2_ atmosphere. Mouse breast cancer 4T1 cells were maintained in RPMI-1640 medium with 10% FBS at 37 °C in a 5% CO_2_ atmosphere. MCF-10A cells were cultured in MCF-10A specific medium at 37 °C in 5% CO_2_ atmosphere.

### 2.7. Cellular Uptake of C210 Prodrug Nanoparticles

The cellular uptake of prodrug nanoparticles was measured using a confocal laser scanning microscope (CLSM) and flow cytometry (FCM). MCF-7 cells were seeded in a culture dish with 1×10^5^ cells and cultured overnight at 37 °C with 5% CO_2_. After that, the medium was taken place of a new one containing free coumarin-6 or coumarin-6-labeled prodrug nanoparticles at the coumarin-6 concentration of 200 ng/mL for 0.5 h or 2 h. Subsequently, the medium was removed and cells were washed with PBS three times. Then, cells were digested by trypsin and suspended in PBS. Cell suspension was analyzed using a FACS Canto (TM) II (Becton, Dickinson and Company, Franklin Lakes, NJ, USA) at an excitation of 488 nm for coumarin-6. After removing the medium, the cells were fixed with 4% paraformaldehyde and stained with Hoechst 33342. Then, the cells were observed using a confocal laser scanning microscope (Leica SP5, Heidelberg, Germany).

### 2.8. Cell Viability Assays

MTT assay was performed to determine the cell viability of MCF-7 cells, 4T1 cells, MCF-10A cells and H22 cells treated with C210 or prodrug nanoparticles. A 96-well plate was used to seed the cancer cells with 3000 cells per well and the cells were cultured for 12 h until they were fully attached. Then, the culture medium was replaced by medium with C210 or prodrug nanoparticles at C210 concentrations ranging from 0.78 to 200 μmol/l. Then, cells were incubated for 48 h or 72 h. After that, 15 μL of MTT solution was added to each well and the cell continued to incubate for 4 h at 37 °C. After the removal of the culture medium, formazan crystals were dissolved with 200 μL DMSO. Then, the absorbance of the plates was measured using a microplate reader at the wavelength of 570 nm.

### 2.9. Animal Studies

A total of 72 female BALB/c mice (5–6 weeks) and 48 female ICR mice (4–5 weeks) were purchased from Shanghai SLAC Laboratory Animal Co., Ltd. A total of 18 female Sprague Dawley (SD) rats (180–200 g) were purchased from Fujian Medical University Laboratory Animal Center. All animal procedures were approved and controlled by the Laboratory Animal Ethics Committee of Fujian Medical University. The animal experiments were carried out according to the guidelines of Chinese law concerning the protection of animal life.

### 2.10. Pharmacokinetic and Biodistribution of C210 Prodrug Nanoparticles

The female SD rats were used to evaluate the pharmacokinetics. The rats were fasted for 12 h with free access to water before experiments. A total of 18 rats were divided into 6 groups and injected intravenously with free C210 or C210 prodrug nanoparticles at an equivalent dose of 20 μmol/kg C210. Blood samples were collected at the predetermined time points (0.08, 0.15, 0.5, 1, 1.5, 2, 4, 6, 8 and 12 h). Then, the plasma was obtained via centrifugation. The concentration of C210 and the corresponding prodrugs was measured via LC-MS. Pharmacokinetic parameters were analyzed using Drug and Statistics 2.0 (DAS) software.

The biodistribution of C210 and the prodrug nanoparticles was investigated by measuring the concentration in the major organs and tumors of 4T1 tumor-bearing BALB/c mice. A total of 72 mice were randomly divided into 6 groups and injected intravenously with free C210 or C210 prodrug nanoparticles at an equivalent dose of 20 μmol/kg C210. After intravenous administration at 1, 2, 4 and 12 h, the mice were sacrificed, and their major organs (including the heart, liver, spleen, lung, kidney and tumor) were collected and then stored at −80 °C until further analysis. Before analysis, tissue samples were thawed to room temperature and homogenized by adding normal saline in the ratio of 1:3 (*w*/*v*). The concentration of C210 and the corresponding prodrugs was determined using the LC-MS method.

### 2.11. Antitumor Effect of C210 Prodrug Nanoparticles In Vivo

The antitumor efficacy of C210 and the prodrug nanoparticles was measured using a mouse breast cancer 4T1 subcutaneous tumor model, whereby 1 × 10^6^ 4T1 cells were injected into the right flank of female BALB/c mice. When the tumor volume reached 100 mm^3^, the bearing tumor mice were randomly assigned into 6 groups (saline, CTX, C210 and C210-S-OA-NPs, C210-SS-OA-NPs and C210-C-OA-NPs) with 8 mice in each group. The treatment of CTX (dose: 30 mg/kg) was administrated to the animals via intraperitoneal injection once every 3 days. Others were intravenously injected once every day (equivalent C210 80 μmol/kg). Mice were sacrificed once they had been treated for 14 days or tumor sizes reached 2000 cm^3^. The major organs and tumors of the mice were collected for TUNEL and H&E for histological examinations. To evaluate hepatorenal function, the serums of mice were collected and tested for alanine transaminase (ALT), aspartate transaminase (AST), creatinine (CRE) and urea nitrogen (BUN). A mouse liver cancer H22 subcutaneous tumor model was established by injecting 1 × 10^6^ H22 cells into the right flank of ICR mice. The treatment regimen was the same as the previous 4T1 tumor models.
Tumor inhibition rate (%) = 1 − (average tumor weight in the treated group)/(average tumor weight in the control group) × 100%(4)

### 2.12. Statistical Analysis

All data were expressed as means ± SD. Statistical analysis was performed using SPSS 22.0 software. Student’s *t*-test and one-way analysis of variance (ANOVA) were used to evaluate the significance. The value of *p* < 0.05 was considered to be statistically significant (*: *p* < 0.05, **: *p* < 0.01, ***: *p* < 0.001).

## 3. Results and Discussion

### 3.1. Synthesis of C210 Prodrugs

Using a single sulfur bond or disulfide bond as the redox-responsive linkages, three C210-OA prodrugs were designed and synthesized, including C210-S-OA containing a single sulfur bond, C210-SS-OA containing a disulfide bond and the control compound C210-C-OA containing only a carbon bond but no redox response bond (Figure 1B). The synthetic routes of the prodrugs are shown in Figure 1A; oleyl alcohols were directly conjugated with linkages to give intermediates S-OA/SS-OA/C-OA with yields of 24%, 26% and 21%, respectively. Then, the intermediates were conjugated with C210 to attain C210-S-OA, C210-SS-OA and C210-C-OA with yields of 63%, 66% and 70%, respectively. Additionally, the chemical structures of these prodrugs were confirmed via the infrared spectrum (IR), high-resolution mass spectrometry (MS), nuclear magnetic resonance hydrogen spectroscopy (^1^H NMR) and nuclear magnetic resonance carbon spectroscopy (^13^C NMR) (Appendix A). The purity of prodrugs was assured via HPLC (Appendix A).

### 3.2. Preparation and Characterization of C210 Prodrug Nanoparticles

We prepared the prodrug self-assembled nanoparticles using a simple precipitation method (Figure 1C). All prodrug preparations could self-assemble to form clear colloidal solutions in aqueous solution, while the same concentration of C210 was precipitated in water under the same conditions (Figure 1A). An obvious Tyndall effect could be observed in the C210 prodrug nanoparticles as the laser could pass through all solutions, except for the free C210 suspension (Figure 1B). This indicated that the modification of C210 via oleyl alcohol could assist in the ability to self-assemble to form colloids in water, which was consistent with the literature [28,29,30]. As shown in Figure 1C,D and Appendix A, these prodrug nanoparticles were spherical in shape with a diameter of about 120 nm and zeta potential of about −35 mv. These prodrug nanoparticles also had a very high drug loading capacity (~50%), thanks to the chemical linkage that allowed the prodrugs themselves to act as both cargo and carrier. These prodrug nanoparticles were incubated with PBS solution containing 10% FBS, and no obvious change was observed either at 37 °C for 48 h or at 4 °C in a dark environment for 3 months (Figure 1E,F), and they exhibited colloidal stability and long-term storage potential.

### 3.3. Redox-Responsive Release of C210 Prodrug Nanoparticles

The release ability of the prodrug nanoparticles was evaluated by simulating a redox environment in vitro. As shown in Figure 2A, when prodrug nanoparticles were incubated in PBS solution (pH 7.4) for 24 h at 37 °C, only a small amount of C210 (less than 10%*w*/*w*) was released. When incubated with PBS containing 1 or 10 mM H_2_O_2_, the prodrug nanoparticles released C210 rapidly, with the release rate in the order of C210-S-OA > C210-SS-OA > C210-C-OA (Figure 2B,C). It was found that the single sulfur bond had a stronger oxidative responsive release than the disulfide bond, while C210-C-OA without a redox-responsive bond had almost no responsive release. In the presence of H_2_O_2_, the sulfur bonds present in the prodrug were oxidized to form hydrophilic sulfoxide groups, which was conducive to the hydrolysis of adjacent ester bonds and the drug release from the prodrug [25]. The oxidation of the disulfide bond required more oxygen to form the sulfone than the single sulfur bond [26]. Moreover, the prodrug nanoparticles also exhibited DTT responsive drug release, and the cumulative release of prodrug nanoparticles followed the order of C210-SS-OA > C210-S-OA > C210-C-OA (Figure 2E,F). The disulfide-bond-containing prodrugs exhibited a more sensitive reduction response, in which C210-SS-OA was fully reduced in about 2 h. In contrast, under non-reducing conditions, the nanoparticles were converted slowly within 24 h (Figure 2D). As shown in Appendix A, the cumulative release behavior of each prodrug nanoparticle in vitro under H_2_O_2_ and DTT conditions appeared to be first-order kinetics. Taken together, both single-sulfur- and disulfide-bonded prodrug nanoparticles had redox-responsive properties, while carbon-bonded prodrug nanoparticles were insensitive to redox environments. Differences in prodrug release may lead to differences in activity.

### 3.4. Cellular Uptake of C210 Prodrug Nanoparticles

Free coumarin-6 or coumarin-6-labeled C210 prodrug nanoparticles were incubated with MCF-7 cells for 0.5 h and 2 h; then, the cellular uptake of the nanoparticles was observed using laser confocal microscopy, and intracellular coumarin-6 fluorescence intensity was quantified via flow cytometry. As shown in Figure 3A–D, the prodrug nanoparticles labeled with coumarin 6 exhibited higher fluorescence intensity than free coumarin-6 at both 0.5 h and 2 h, which revealed that the prodrug nanoparticles had higher cellular uptake efficiency. The cellular uptakes among the different prodrug nanoparticles were not significantly different, probably due to their similar physiochemical properties thus making them possess close uptake rates [31].

### 3.5. Cytotoxicity of C210 Prodrug Nanoparticles on Cancer Cells

Since there was no difference in cellular uptake of prodrug nanoparticles, we hypothesized that the cytotoxicity of prodrug nanoparticles depended on their intracellular release capacity. Cell viability was tested via MTT assay to determine the effect of C210 and the prodrug nanoparticles on cancer cells. As shown in Figure 4A–D and Appendix A and Appendix A, C210-S-OA NPs and C210-SS-OA NPs showed potent antitumor activity in breast cancer MCF-7, 4T1 cells and liver cancer H22 cells, while C210-C-OA NPs had the lowest antitumor activity. Corresponding with this, C210-S-OA NPs and C210-SS-OA NPs could release more C210 than C210-C-OA NPs in MCF-7 cells (Appendix A). The results indicated that the antitumor activity of the prodrug nanoparticles in vitro was dependent on their release of C210. To compare the selectivity of C210 prodrug nanoparticles and C210 on cancer cells, we observed the cytotoxicity of the prodrug nanoparticles and their primary drug on breast cancer cells (MCF-7) and normal breast epithelial cells (MCF-10A), then calculated the relative therapeutic index (IC_50 normal cell_/IC_50 cancer cell_). It was found that the cytotoxicity of C210-S-OA and C210-SS-OA to normal cell (MCF-10A) was much lower than that of tumor cell (MCF-7), and the relative therapeutic index was higher than that of C210 (Appendix A). The results indicated that the redox-responsive C210 prodrug nanoparticles had higher selectivity on cancer cells than C210. This may be due to the different redox backgrounds of cancer cells and normal cells (Appendix A). Meanwhile, we found that DSPE-PEG 2000 and OA had no significant cytotoxicity on MCF-7 cell concentrations at 200 μg/mL, which was much higher than the highest concentration possible used in the experiments (Appendix A). This indicated that the cytotoxicity of C210 prodrug nanoparticles came from C210.

Similar to their effect on cancer cell viability, the redox-responsive C210 prodrug nanoparticles were able to induce the apoptosis of cancer cells. As shown in Figure 4E and Appendix A, regarding MCF-7 cells treated with the nanoparticles for 48 h or 72 h, C210-S-OA NPs and C210-SS-OA NPs could significantly induce apoptosis, while C210-C-OA NPs showed almost no activity for inducing apoptosis.

### 3.6. Pharmacokinetics and Biodistribution of C210 Prodrug Nanoparticles

The pharmacokinetic profiles of C210 prodrug nanoparticles and C210 released from them were evaluated after the administration of the prodrug nanoparticles via the tail vein in SD rats. The mean plasma concentration–time profiles are shown in Figure 5A,B, and the main pharmacokinetic parameters are listed in Appendix A; it was found that free C210 was rapidly eliminated from the blood circulation. In contrast, the mean retention time (MRT) of released C210 from the prodrug nanoparticles in blood circulation was significantly prolonged. The MRTs of C210-S-OA NPs, C210-SS-OA NPs and C210-C-OA NPs were 7.3, 6.8 and 5.7 times that of the free C210 group, respectively. The AUC of C210 released from the redox-responsive C210 prodrug nanoparticles increased significantly compared to free C210, i.e., the AUCs of C210-S-OA NPs, C210-SS-OA NPs and α-C210-C-OA NPs were 9.9, 9.6 and 0.6 folds that of free C210, respectively. These results suggest that C210-S-OA NPs and C210-SS-OA NPs showed favorable pharmacokinetic behaviors compared to C210. C210-C-OA NPs could not be released quickly in vivo and could only prolong the systemic circulation time, but could not improve bioavailability. Taken together, these show that prodrugs modified with oleic acid can effectively prolong the systemic circulation time, while the quick release of prodrugs in vivo is required to improve bioavailability.

4T1 tumor-bearing mice were used to study the biodistribution of the nanoparticles (Figure 5C–E and Appendix A). Consistent with their pharmacokinetic results, the accumulation of C210 prodrug nanoparticles in the major organs was much higher than that of the free C210. The prodrug nanoparticles were distributed mainly in the liver and spleen. C210-S-OA NPs released less C210 than C210-SS-OA NPs in various organs, but more in tumor tissue, which means that C210-S-OA NPs might have lower potential toxicity in normal tissues and higher selective toxicity in tumor tissue. The accumulation (AUC_0~12 h_) of C210 released from C210-S-OA NPs, C210-SS-OA NPs and C210-C-OA NPs in tumor tissue was 3.3, 1.5 and 0.3 folds that of free C210, respectively (Appendix A). Additionally, the duration of C210 released from C210-S-OA NPs in tumor tissue was much longer than that of free C210 (Figure 5F). This suggests that the redox-responsive nano-delivery system could effectively increase the exposure of C210 in tumor tissue. The results demonstrate that oleyl alcohol and redox-responsive bonds in these prodrugs significantly improved their pharmacokinetic profiles and enhanced the tumor-specific exposure of C210, which might be the prerequisites for effectively killing tumors.

### 3.7. Antitumor Effects of C210 Prodrug Nanoparticles In Vivo

Mouse breast cancer 4T1 cell and liver cancer H22 cell subcutaneous inoculation models were used to evaluate the antitumor effects of the C210 prodrug nanoparticles in vivo. It was found that for both 4T1 and H22 tumor-bearing mice, the tumor volumes in the treatment groups were inhibited during the treatment period and the growth inhibition in the C210-S-OA and CTX groups was the most remarkable (Figure 6A,B and Figure 7A,B). At the end of the treatment, the tumor weight inhibition rates of the CTX, C210, C210-S-OA, C210-SS-OA and C210-C-OA groups in the 4T1 tumor-bearing mice were 60.6%, 49.7%, 63.9%, 46.2% and 27.6%, respectively (Figure 6C). Then, 4T1 tumor tissues were collected to test the apoptosis induced by treatment, and the results showed that extensive apoptosis cancer cells were found in the tumor tissues of the CTX and C210-S-OA NP groups (Appendix A). Similarly, the tumor weight inhibition rates of the CTX, C210, C210-S-OA, C210-SS-OA and C210-C-OA groups in the H22 tumor-bearing mice were 69.5%, 44.1%, 74.8%, 44.5% and 45.3%, respectively (Figure 7C). The results indicated that the C210-S-OA NPs, the prodrug nanoparticles containing the single sulfur bond, had the most potent antitumor effect in vivo, and their antitumor effect was stronger than C210 without formulation, even stronger than the positive control cyclophosphamide (CTX). C210-C-OA NPs exhibited a poor antitumor effect mainly due to their inability to rapidly release C210 and lower distribution in tumor tissue. The antitumor activity of the C210-S-OA NPs was stronger than the C210-SS-OA NPs, which implies that it might be easier to release C210 in tumors using the monosulfide C210 prodrug than the disulfide prodrug. This result was consistent with that of Liang [32], while Sun [26] and Wang [33] suggested that the disulfide bond was more effective in vivo. This may be due to different parent drugs and/or different intracellular cancer cell redox levels.

In addition, we evaluated the biosafety of the prodrug nanoparticles in mice during the whole treatment period. As shown in Figure 6D and Figure 7D, no significant change in body weight was found in groups treated with C210 prodrug nanoparticles compared with the group treated with saline. At the end of the treatment, neither C210 nor its prodrug nanoparticles were significantly changed in the liver and kidney function of mice (Appendix A). Consistent with this result, no significant change was found in the HE staining results of the major organs of mice (Appendix A). The above data demonstrated that C210 and its prodrug nanoparticles within their effective dose ranges had no toxicity and a good safety profile.

## 4. Conclusions

We successfully synthesized the redox response C210 prodrugs containing oleyl alcohols and single sulfur or disulfide bonds, and prepared them into a self-assembled nano-delivery system with a high drug loading capacity, high water solubility and good stability. Among them, the prodrug nanoparticles containing a single sulfur bond (C210-S-OA NPs) were the most sensitive to the intracellular redox of cancer cells. They could release C210 rapidly in cancer cells, showing favorable pharmacokinetic properties, and had stronger antitumor activity in vivo than C210 or other prodrug nanoparticles. In conclusion, the redox-responsive lipidic prodrug nano-delivery system could improve the antitumor effect of the curcumin derivative C210 in vivo, providing a basis for the further clinical translation of curcumin and its derivatives.

## Data Availability

Data are contained within the article or Appendix A. The data presented in this study are available upon request from the corresponding author.

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
