# Peer review of "Redox-Responsive Lipidic Prodrug Nano-Delivery System Improves Antitumor Effect of Curcumin Derivative C210"

_pharmaceutics, 2023, doi:10.3390/pharmaceutics15051546_

Round 1

Reviewer 1 Report

This manuscript synthesized three conjugates of C210 and oleyl alcohol (OA) via different linkages containing single sulfur/disulfide/carbon bonds to improve the bioavailability antitumor activity of curcumin derivative C210. The prodrug nanoparticles (C210- S-OA NPs) dramatically improved their pharmacokinetic behavior, were sensitive to the intracellular redox level of cancer cells, and rapidly released C210 in cancer cells, inducing the strongest cytotoxicity to cancer cells and antitumor activity in the mouse model. The author comprehensively tested and analyzed the in vivo and in vitro properties of nanoparticles.

1. In Figure 1C, some nanoparticles appear to form vesicle structures, please explain the reason for this phenomenon.

2. The author used DTT and H2O2 to test the drug release under redox conditions while obtaining different release curves. Under different conditions, the response results of single-sulfur and disulfide-bonded prodrug nanoparticles are completely different. What is the reason for this? Although DTT and H2O2 are commonly used redox reagents, which is more similar to the intracellular redox situation?

3. In Figure 3B, the cellular uptake of C210-S-OA NPs and C210-C-OA NPs looks higher than C210-SS-OA NPs. Please retest the cellular uptake by Flow and show the MFI of each group.

Reviewer 2 Report

The manuscript pharmaceutics-2368317 "Redox responsive lipidic prodrug nano-delivery system improves antitumor effect of curcumin derivative C210" by Guo et al. is describes the synthesis of novel redox responsive conjugates based on curcumin derivative C210 and oleyl alcohol with different linkers containing single sulfur/disulfide/carbon bonds, the preparation of nanoparticles based on the obtained compounds and the study of their cytotoxicity and antitumor potential. The authors obtained interesting SARs, so I think this paper will be of interest to the readers of Pharmaceutics.

However, I have some questions and comments.

1) There are some problems with superscripts and subscripts in the text of the manuscript and supplementary materials (1H NMR, C55H74O13S2-H, CDCl3 etс.).

2) Lines 63-64. References "Feng et al., 2019; Luo et
al., 2016a; Sun et al., 2019" should be given according journal’s rules (i.e., [1,2,3]).

3) Please add the yields of the target compounds in grams and percentages.

4) Scheme 1, part A. The use of R1, R2, R3 is incorrect, since there is only R in the scheme.

5) Please add DTT abbreviation deciphering.

6) How are coumarin-6 labeled prodrugs nanoparticles synthesized?

7) How can the authors explain the almost 2 orders of magnitude decrease in antitumor activity upon going from C210 to nanoparticles (C210-S-OA and C210-SS-OA)?

8) Please add a comparison of the results obtained with the literature, if any.

Reviewer 3 Report

This study deals with the design and the development of Redox responsive lipidic prodrug nano-delivery system of curcumin derivative C210 . This investigation includes a gamut of techniques for the evaluation of the final formulation.

The maunsript is well organized and well-written.

My comments are:

1. The authors should discuss in depth the physicochemical characteristics of the formulation. Why the zeta potential is positive and it role to the cellular uptake.

2. The Kinetics (first -order etc.) of the release profile should be included in the discuss.

3. Why the nanoprecipitation method was used?

 Minor editing of English language required

Reviewer 4 Report

I have read the paper with great interest since the NPs formulation of curcumin derivatives is an appreciable strategy to improve the bioavailability for exploiting the pharmacological profile of curcuminoids. The paper is well written and the experiments were conducted with competency. I appreciate the work. Accordingly, I have only minor comments in order to give some details to the readers.

-purity of purchased compounds should be reported in materials and methods section according to vendors information

-number of animals should be reported

-it is required to report the IC50 in the cytotoxicity studies (paragraph 3.5)

-it is not clear the administered dose of the NPs in vivo and the data should be included in the text.

recent literature in the field should be cited Tabanelli et al 2021 Improving curcumin biovalability:current strategies and future perspective https://doi.org/10.3390/pharmaceutics13101715

the english language is fine.

Round 2

Reviewer 2 Report

I thank the authors for answering my questions and improving the manuscript